# Effects of Different Levels of Green Tea Powder on Performance, Antioxidant Activity, Egg Mass, Quality, and Cecal Microflora of Chickens

**DOI:** 10.3390/ani14203020

**Published:** 2024-10-18

**Authors:** Wei Luo, Qisong Tan, Hui Li, Tao Ye, Tao Xiao, Xingzhou Tian, Weiwei Wang

**Affiliations:** 1Key Laboratory of Animal Genetics, Breeding and Reproduction in the Plateau Mountainous Region, Ministry of Education, Guizhou University, Guiyang 550025, China; lw2023202303@163.com (W.L.); gzutqs@163.com (Q.T.); yetao1416979455@sina.com (T.Y.); xaoao1995@163.com (T.X.); tianxingzhou@yeah.net (X.T.); wangww14@lzu.edu.cn (W.W.); 2College of Animal Science, Guizhou University, Guiyang 550025, China; 3Weining County Animal Disease Prevention and Control Center, Bijie 553100, China

**Keywords:** *green tea powder*, laying hens, egg quality, antioxidant index, cecal microorganism

## Abstract

The addition of alternative feed sources such as *Green Tea Powder* to the feed of poultry, especially laying hens, has attracted attention due to its potential benefits. Therefore, the purpose of this study—with the addition of *Green Tea Powder* to the diet of laying hens—is, on the one hand, to explore the beneficial effects of *Green Tea Powder* on the production performance and physical function of laying hens and, on the other hand, to explore the utilization of *Green Tea Powder* and improve the utilization rate of biological resources. In conclusion, dietary supplementation of *Green Tea Powder* during the laying period can reduce the serum yolk weight of laying hens. The serum levels of immunoglobulin A and immunoglobulin M were increased, and the serum levels of malondialdehyde were decreased. Egg shell strength and shell thickness decreased, and egg yolk color deepened. Additionally, the percentage of amino acids in the eggs increased. It is beneficial to the health of laying hens to increase the diversity of intestinal flora and improve the structure of cecum flora. This study investigated the effects of *Green Tea Powder* supplementation on the performance and egg quality of laying hens and ultimately guided the development of more effective and scientific animal feeding strategies.

## 1. Introduction

China, a country with a high tea production and consumption, produces a large amount of tea leaves every year, which not only pollutes the environment but also causes a substantial waste of biological resources [1]. Studies have found that green tea is rich in various bioactive ingredients such as polyphenols, vitamins, saponins, and trace elements, which can enhance the antioxidant capacity of the animal body, improve bacteriostasis and disease resistance, and modulate the regulation of animal immunity and other functions [2]. In addition, green tea has anticancer and blood fat-lowering properties [3], and the large number of active ingredients in tea has attracted interest from researchers.

Eggs are a staple food in the human diet, and with the continuous improvement in the living standards of consumers, the quality of eggs is receiving increasing attention [4]. The color, taste, shell, and color of egg yolk affect a consumer’s purchase desire; therefore, the healthy breeding of laying hens is important [5]. Among others factors, the laying performance of hens is affected by genetic potential, nutrition, and feeding management. Intestinal health plays an important role in the growth and production of poultry, and their intestinal tract, the largest immune organ in their body, is the key component of nutrient digestion and absorption [6].

Green tea can inhibit many harmful bacteria such as *Escherichia coli*, *Staphylococcus aureus*, and *Clostridium percapsulens* [7,8]. Polyphenols and epigallocatechin gallate (EGCG) in green tea can improve the growth performance and egg quality of laying hens and regulate their intestinal health [9,10]. Dietary polyphenols can maintain the balance of intestinal microbes by stimulating the growth of beneficial bacteria and inhibiting that of the pathogenic bacteria, thus contributing to the maintenance of intestinal health [11,12]. Based on the results of previous studies, we hypothesized that green tea residue can improve the egg quality, immune performance, and antioxidant performance of laying hens and regulate the structure of their cecal microflora. The purpose of this study was to examine the effects of adding different proportions of *Green Tea Powder* on the performance, egg quality, serum antioxidant and immune indices, and intestinal microorganisms of laying hens during the peak laying period, in order to provide a theoretical reference for the rational utilization of a *Green Tea Powder* supplement in the feed of laying hens.

The beneficial effects of adding *Green Tea Powder* to the diet of laying hens are as follows: On the one hand, green tea and its by-products can be developed as feed additives for reducing the use of antibiotics, and therefore it is a good antibiotic substitute. It improves the safety of food and reduces breeding costs. On the other hand, it can ensure that biological resources such as green tea by-products are more fully utilized, avoid environmental pollution, reduce the waste of resources, and increase the economic income of the country.

## 2. Materials and Methods

This study was approved by the Animal Welfare Committee of Guizhou University, Guiyang, China (Approval number:EAE-GZU-2020-P019).

### 2.1. Green Tea Powder, Laying Hens, Feeding, and Experimental Design

The breeding experiment was carried out at the Guizhou Zhuxiang Chicken Breeding Co., Ltd. (location: Chishui, China, coordinates: 105.756466° N, 28.583474° E). A total of 360 Chishui black-bone chickens (42 to 43 weeks of age) with a body weight of 1711.34 ± 96.01 g were selected and randomly allocated into four groups using a single factor and completely randomized trial design. The treatment groups were as follows: (1) control, basal diet without *Green Tea Powder* supplementation; (2) trial group I, basal diet + 0.8% *Green Tea Powder*; (3) trial group II, basal diet + 1.6% *Green Tea Powder*; and (4) trial group III, basal diet + 2.4% Green Tea Powder. Each group consisted of six replicates with 15 hens per replicate. Each replicate was further divided into five cages, with three laying hens in each cage that were given access to a free choice diet and drinking water. The experimental temperature was 23 °C, the humidity was 60–65%, the area was illuminated for 16 h daily, the pre-experimental period lasted for 14 d, and the experimental period had a duration of 60 d. The nutritional requirements of the laying hens were met in accordance with the National Research Council (NRC) guidelines [13]. The *Green Tea Powder* used in the study was Cuiya tea from Meitan, Guizhou, purchased from Zunyi Tea (Group) Co., Ltd. (Zunyi, China), and it was added to the diet of the laying hens in a step-by-step mixing method. The feed formula and nutrient levels are shown in Table 1.

### 2.2. Sample and Measurements

**Nutrient Composition of the Diet:** In total, 500 g of each experimental diet was collected and dried in an oven at 65 °C by the quartering method, moisture was restored at room temperature (20 °C to 25 °C) for 24 h, and the feed was crushed through a 1 mm sieve and transferred into a self-sealing bag for air drying. The dietary dry matter (DM), crude protein (CP), ether extract (EE), ash, calcium (Ca), and phosphorus (P) contents were determined according to international standards of the Association of Official Analytical Chemists (AOAC) [14], shown in Table 1. The content determination of the functional components in *Green Tea Powder* refers to the method of Wang [15] and the results are shown in Table 2.

**Production Performance:** The body weight in the fasting state of the laying hens was measured at 5 am. on the first and last day of the experiment, representing the initial and final body weight, respectively. During the experiment, feed intake, egg production, and egg weight were recorded in all groups. The average daily gain (ADG), average daily feed intake (ADFI), egg production (EP), and feed conversion rate (FCR) were calculated after the completion of experiments.

**Determination of Serum Immune and Antioxidant Indices:** At the end of the feeding experiment, two hens were randomly selected from each replicate, for a total of 48 hens. First, 5 mL blood was collected from the hen’s subwing vein using a normal tube collection vessel, and the tube was placed in an ice bag. Then, the blood samples were centrifuged (Heraeus Multtfuge XIR, Thermo, Waltham, MA, USA) at 3000 rpm for 15 min. The serum was divided into sterile centrifuge tubes, sealed, and stored at −20 °C. The serum levels of immunoglobulin A (IgA), immunoglobulin G (IgG), and immunoglobulin M (IgM) were determined by enzyme-linked immunosorbent assay (ELISA). The kits were purchased from the Nanjing Jiancheng Institute of Biological Engineering, and the item numbers were H108-1-1, H106-1-1, and H109-1-1, respectively. The levels of superoxide dismutase (SOD), glutathione peroxidase (GSH-Px), malondialdehyde (MDA), total protein (TP), and albumin (ALB) in the serum were determined by a colorimetric method, and the kit was purchased from the Nanjing Jiancheng Institute of Biological Engineering, with item numbers A001-2-2, A005-1-2, A003-1-1, A045-4-2, and A028-1-1, respectively. The indices were measured in accordance with the kit operation procedures, and the instrument used was a Power Wave XS full wavelength microplate analyzer (Bio-Tek Instruments Inc., Winooski, VT, USA).

**Egg Quality:** At the end of the feeding experiment, 72 eggs (3 eggs per replicate) were collected, and the egg quality was determined within 24 h. The average egg weight (AEW), yolk color (YC), albumen height (AH), and Haugh unit (HU) for all eggs were measured using an egg quality tester (EA-01, Orka, Ramat Hasharon, Israel). YC was analyzed by using a Minolta Chroma Meter CR-300 (Minolta Co. Ltd., Osaka, Japan), where the L*, a*, and b* values were recorded and reflected lightness (0 = black, 100 = white), redness (−100 = green, 100 = red), and yellowness (−100 = blue, 100 = yellow), respectively. The eggshell strength (ES) was measured using an eggshell strength tester (EFA-01, Orka, Israel), and the unit used was Nm^−2^. The egg yolk was weighed using an electronic balance (BSA224S, Sartorius Scientific Instruments Co., Ltd., Beijing, China) to the nearest 0.01 g. The eggshell thickness (ET), lateral diameter, and longitudinal diameter in three eggshell zones (apical, equatorial, and basal) were measured using a digital vernier caliper (MNT-150T Shanghai Meinet Industrial Co., Ltd., Shanghai, China), accurate to 0.01 mm. The ET is the average of the three eggshell zones, and the egg shape index (ESI) is the transverse diameter divided by the vertical diameter.

**Amino Acid and Cholesterol Content Detection:** On the last day of the experiment, 3 eggs were collected per replicate (72 eggs in total). The contents of amino acids and cholesterol in the whole eggs were determined according to the methods in GB 5009.124-2016 (https://max.book118.com/html/2019/0805/8046134143002040.shtm, accessed on 14 October 2024) and GB 5009.128-2016(https://www.renrendoc.com/paper/239476492.html, accessed on 14 October 2024). The amino acids in the feed were determined using the same method as that used for eggs. The main instruments used in the test were a high-performance liquid chromatograph (1268 DBAB805075, Agilent Technology Co., Ltd., Beijing, China) and an automatic amino acid analyzer (L-8900 Manufacturing Co., Ltd., Tokyo, Japan).

The following steps were performed to determine the amino acid content of the eggs: Sample preparation: the eggshell was removed and the eggs were homogenized. Sample weighing: The weighed sample was placed in a hydrolysis tube (accurate to 0.0001 g), ensuring that the protein content of the sample was between 10 and 20 mg. If the protein content of the sample was unknown, the protein content of the sample was measured first. Sample hydrolysis: Depending on the protein content of the sample, 10~15 mL of 6 mol L^−1^ hydrochloric acid solution and 3~4 drops of phenol were added to the hydrolysis tube. The hydrolysis tube was placed into a refrigerator for 3~5 min, vacuumed, and filled with nitrogen. This process was repeated three times before sealing the tube. The hydrolysis tube was placed in an electric blast thermostat for hydrolysis at 110 ± 1 °C for 22 h and then removed and cooled to a constant temperature. The hydrolysate was filtered into a 50 mL volumetric flask. The filtrate (1.0 mL) was placed in a 15 mL test tube and dried under reduced pressure at 40~50 °C using a parallel evaporator. The residue was dissolved in 1~2 mL water, and the pressure was then reduced. The residue was dried and finally evaporated. The solution was dissolved with 1~2 mL of pH 2.2 sodium citrate buffer and transferred through a 0.22 μm filter to the injection flask of the instrument for determination. Determination: The standard working solution of the mixed amino acids was injected into an amino acid automatic analyzer, and the operating procedures, parameters, and reagent ratio of the buffer solution for elution were adjusted according to the verification regulations of the JJG 1064-2011 amino acid analyzer and the instrument manual. The same volumes of the standard working solution for mixed amino acids and sample determination solution were injected into the amino acid analyzer. The concentration of the amino acids in the sample determination solution was calculated using the external standard method based on the peak area.

**Chromatographic Conditions:** the standard analysis column was 4.6 mm × 60.0 mm, the separation column temperature was 57.0 °C, and the reaction column temperature was 136 °C.

For the determination of cholesterol in the eggs, the following steps were performed: Sample preparation: the eggs were shelled and homogenized. Chromatographic conditions: The capillary column SH-RT-2560 was 100 m × 0.25 µm × 0.20 µm. For the heating procedure, the initial temperature was 100 °C, which was maintained for 13 min, then increased to 180 °C at 10 °C min^−1^ for 8 min and 1 °C min^–1^ for 20 min, and increased to 230 °C at 4 °C min^–1^ for 12 min. The flow rate of the carrier gas (high-purity helium gas) was set at 1.0 mL min^–1^.

**Cecal Flora Determination:** On the last day of the feeding experiment, one hen was randomly selected from each replicate and euthanized, for a total of 24 chickens. The cecal contents were collected under aseptic conditions and stored in freezing tubes at −80 °C. Then, the tubes were sent to Shanghai Meji Biomedical Technology Co., Ltd.(Shanghai, China) to sequence the bacterial 16S rRNA using the MiSeq platform. The sequencing process used is as follows: The total bacterial DNA from the cecal contents was extracted by referring to the procedures of the QIAamp Fast DNA Stool Mini Kit (D4015, Omega, Inc., Norwalk, CT, USA). The purity and concentration of DNA were detected using a Nanodrop spectrophotometer (Nanodrop Technologies, Wilmington, DE, USA). The DNA integrity was detected using 1% agarose gel electrophoresis, and DNA with an OD260/OD280 ratio within 1.8~2.0 was qualified. The primers were synthesized according to the full-length 16S rRNA primers 338 F and 806R (338F: 5-ACTCCTACGGGAGGCAGCAG-3; 806R: 5-GGACTACHVGGGTWTCTAAT-3). TransGen AP221-02: Trans start FastPfu DNA polymerase and a 20 µL reaction system were used for the polymerase chain reaction (PCR) amplification (5 × FastPfu buffer [4 µL], 2.5 mM deoxyribonucleotide triphosphates [dNTPs; 2 µL], 5 µM forward primer [0.8 µL], 5 µM reverse primer [0.8 µL], 0.4 µL of FastPfu polymerase, 0.2 µL of bovine serum albumin [BSA], 10 ng template DNA, and 20 µL of ddH_2_O). The temperature sequence used was 1 × (3 min at 95 °C), 27 × (30 s at 95 °C, 30 s at 55 °C, 45 s at 72 °C), 10 min at 72 °C, and 10 °C until halted by the user. The PCR products were detected by 2% agarose gel electrophoresis, and the products were inspected to ensure that the target band size of the PCR products was correct, the concentration was appropriate, and subsequent tests could be carried out. After passing the quality inspection, the sequencing library was inspected, and high-throughput sequencing was carried out after passing the quality inspection. The paired end reads obtained by MiSeq sequencing were first stitched according to the overlap relationship, and the sequence quality was controlled and filtered. Operational taxonomic unit (OTU) cluster analysis and species taxonomic analysis were performed after the samples were distinguished. Based on the results of the OTU cluster analysis, the diversity index of the OTUs could be analyzed, and the sequencing depth could be detected. Based on the taxonomic information, the community structure could be statistically analyzed at each taxonomic level. Based on the above analysis, a series of indepth statistical and visual analyses such as multivariate analysis and least significant difference tests could be performed on the community composition and phylogenetic information of multiple species.

### 2.3. Statistical Analysis

The test data were recorded and sorted using Excel 2010, SPSS18.0 (SPSS software for Windows; Statistical software SPSS Inc., Chicago, IL, USA) was used to perform a one-way analysis of variance (ANOVA) on the experimental data, and Duncan’s method was used to perform multiple comparisons among groups. Data were presented as “mean ± standard deviation (SD)”, and *p* < 0.05 indicated a significant difference. The 16S rRNA data were analyzed by QIME19.1.1, and the alpha diversity, principal component analysis (PCA), species relative abundance at the phylum and family levels, phylum heat map, and family heat map were measured using the Phyloseq method.

## 3. Results

### 3.1. Production Performance

As shown in Table 3, compared with the control group, the addition of 1.6% and 2.4% *Green Tea Powder* significantly reduced the ADG (*p* < 0.05). Compared with the control group, the effects of different proportions of *Green Tea Powder* on ADFI, EP, and FCR were not significant (*p* > 0.05).

### 3.2. Serum Biochemical Indices, Antioxidant Status, and Immune Status

As shown in Table 4, the levels of MDA and GSH-Px in the serum were significantly decreased as an effect of *Green Tea Powder* (with the exception of group I for GSH-Px) (*p* < 0.05), with no significant effects on the SOD, TP, IgG, and ALB contents (*p* > 0.05). The supplementation level of 0.8% significantly increased the levels of IgM and IgA (*p* < 0.05).

### 3.3. Egg Quality

As shown in Table 5, compared with the control group, the addition of 1.6% *Green Tea Powder* significantly increased the YC and significantly decreased ES and EYW (*p* < 0.05). Compared with the control group, the effects of different proportions of *Green Tea Powder* on AEW, AH, HU, and ESI were not significant (*p* > 0.05). It should be emphasized that eggshell thickness (ET) values were decreased in all experimental groups when compared with the control group.

### 3.4. Amino Acids and Cholesterol in Eggs

As shown in Table 6, the ratio of essential amino acids to total amino acids (EAA/TAA) in all treatment groups ranged from 41.37% to 43.05%, which was higher than the 40% prescribed by the World Health Organization (WHO)/Food and Agriculture Organization (FAO) for ideal proteins. The ratio of essential amino acids to nonessential amino acids (EAA/NEAA) ranged from 70.57% to 75.58%, which was 60% higher than the ideal protein stipulated by the WHO/FAO.

Compared with the control group, dietary *Green Tea Powder* had no significant effect on the cholesterol content in eggs (*p* > 0.05) but had a significant effect on the amino acid content. The addition of 2.4% *Green Tea Powder* significantly increased the contents of TAA, EAA, NEAA, and umami amino acids in eggs (*p* < 0.05), and the amino acid composition ratio became closer to the ideal protein ratio recommended by the WHO/FAO. Compared with the control group, adding 2.4% *Green Tea Powder* had no significant effect on the contents of Met, Tyr, and His in eggs (*p* > 0.05).

### 3.5. Status of Cecal Microflora of Laying Hens

**OTU Statistics and Alpha Diversity Analysis:** As shown in Table 7, a total of 919,477 sequences were obtained by 16S rRNA sequencing from 24 samples of the cecal content of laying hens and counting the number of reads in each sample. The number of effective sequences in all treatment groups was high, ranging from 36,046 to 40,932. The effective sequences were those that contained specific amplification primers without fuzzy bases and that were longer than the standard for analysis. The number of effective sequences in trial group II was significantly higher than those in the control group and trial group I (*p* < 0.05) and not significantly different from that in trial group III (*p* > 0.05).

The number of OTUs in each treatment group fluctuated between 1027 and 1071, and there was no significant difference among the groups (*p* > 0.05). Based on a sequence similarity greater than 97%, the coverage values of the generated OTUs were all greater than 99.7%, indicating good sequencing coverage. There was no significant difference in the Shannon, Simpson, Chao1 abundance-based coverage estimator (ACE), and Chao1 alpha diversity indices among all treatment groups (*p* > 0.05).

As can be observed from the depth analysis curve of the cecal microflora sequence (Figure 1), the Shannon diversity index curve of the cecal microflora in the four treatment groups gradually flattened as the number of sequencing lines increased, indicating that the sequencing covered the information of all samples in each treatment group. Thus, it could accurately reflect the sequence information of the cecal microflora in the samples.

**Venn Diagram Analysis:** Venn diagrams are often used to count the number of unique and shared OTUs in multiple samples. The different treatment groups are represented by graphs of different colors. The overlap between different colored graphs is the number of OTUs shared between the two treatment groups. Venn diagrams can intuitively show the number composition and overlapping OTUs in each sample. As shown in Figure 2, the total number of OTUs in the four treatment groups was 859, and there were 1071 OTUs in the control group (CON), among which 46 OTUs were unique. There were 1027 OTUs in the T1 group, including 6 unique OTUs. There were 1060 OTUs in the T2 group, including 11 unique OTUs. There were 1057 OTUs in the T3 group, of which 18 were unique.

**PCA:** PCA is an algorithm commonly used in statistics. It uses variance decomposition to reflect the differences between multiple groups of data on a two dimensional coordinate graph. The two eigenvalues that best reflect the differences between samples can be selected from the coordinate axes. The closer the distance reflected in the PCA diagram, the more similar the sample species composition. The first principal component (PC1) refers to a set of variables with the greatest variance, and the second principal component (PC2) refers to a set of variables with the second greatest variance. PCA and mapping in R language (Version 3.3.1) were used for 24 samples at the OTU level. As shown in Figure 3, PC1 accounted for 30.06% of the variance, and PC2 accounted for 17.69%. Samples from the same diet treatment group were clustered together. Among them, the CON group had a higher degree of aggregation, and the T2 group had a lower degree of aggregation. The samples in the other groups were relatively distant.

Analysis of Cecal Microfloral Composition and Community Structure: As shown in Figure 4A and Table 8, at the phylum level, seven phyla were detected in addition to unknown flora. *Bacteroides* and *Firmicutes* were the dominant phyla in the four treatment groups. After the difference analysis of the dominant flora among all treatment groups, it was found that the relative abundance of *Bacteroidetes* in the T1, T2, and T3 groups was higher than that in the CON group, and the relative abundance of *Firmicutes* was lower than that in the CON group. The addition of 0.8% *Green Tea Powder* significantly increased the amount of *Bacteroidetes* and decreased the amount of *Firmicutes* (*p* < 0.05). The cluster analysis of the species abundance is shown in Figure 4B. The addition of *Green Tea Powder* increased the relative abundance of *Bacteroidetes* and *Spirochaetota,* while decreasing the relative abundance of *Firmicutes* and *Desulfobacteriota*, thereby changing the structure of flora at the phylum level.

As shown in Figure 4C and Table 9, a total of 16 bacterial families were detected, except for the unknown flora, and *Bacteroidaceae* and *Lachnospiraceae* were the dominant bacterial families in the four treatment groups. Through the difference analysis of the dominant flora among all treatment groups, it was found that the relative abundance of *Bacteroidaceae* in the T1, T2, and T3 groups was higher than that in the CON group, and the relative abundance of *Lachnospiraceae* was significantly lower than that in the CON group. The results indicated that the addition of *Green Tea Powder* increased the abundance of *Bacteroidaceae* and significantly reduced the amount of *Piloricillaceae* (*p* < 0.05).

The cluster analysis of species abundance is shown in Figure 4D. The addition of *Green Tea Powder* increased the relative abundance of *Bacteroidaceae* and *Rikenellaceae.* It also decreased the relative abundance of *Rikenellaceae,* thereby changing the microfloral structure at the family level.

## 4. Discussion

### 4.1. Production Performance

According to a study by Hrnčá et al. [16], feeding different doses of green tea powder resulted in growth inhibition of broilers and significantly reduced the breast muscle and abdominal fat content. This is because the catechins in green tea can inhibit the absorption of intestinal lipids, thereby reducing the fat content in the abdomen. This mechanism may occur owing to the alteration of the formation of micelles that mediate bile acid reabsorption [17]. Murase et al. [18] and Amemiya-Kudo et al. [19] also reported that catechins may increase energy expenditure by stimulating oxidative activity in the liver, thereby reducing fat deposition.

Friedrich et al. [20] showed that EGCG significantly reduces the activity of acetyl-CoA (ACC). Yeh et al. [21] also showed that EGCG in catechin reduced the DNA binding capacity of the promoter-specific transcription factor SP-1 in MCF-7 mammary cells. SP-1 plays an important role in the activation of target genes by sterol response element binding protein (SREBP-1c), which regulates fatty acid synthesis by binding to sterol regulatory elements (SREs) [22]. Therefore, it is speculated that EGCG in green tea leaves can reduce the DNA binding ability of SP-1, thereby weakening the activation of SREBP-1c on fatty acid synthetase mRNA expression and reducing the activity of ACC to inhibit fatty acid synthesis, reduce the deposition of fat, and control growth and body weight.

Dietary tea polyphenols significantly decrease the body weight gain of broilers [23]. Epigallocatechin gallate supplementation significantly increases the feed intake of and egg production in quails [24,25]. The results of our experiment also showed that dietary supplementation with *Green Tea Powder* at the level of 1.6–2.4% significantly reduced the average daily weight gain of laying hens, without affecting egg production and feed intake. Hens in the laying period need to have controlled weight growth.

### 4.2. Serum Biochemical Indices, Antioxidant Status, and Immune Status

The health status and metabolism of poultry can be evaluated by analyzing serum biochemical parameters [26]. In this experiment, the addition of *Green Tea Powder* significantly reduced the MDA content in the serum of laying hens. Adding 1.6% and 2.4% green tea leaves significantly reduced the serum GSH-Px content. Studies have shown that EGCG can regulate the activity of antioxidant enzymes, which can interact with the NF-E2-related factor 2 (NRF2) and Kelch-like ech-associated protein 1 (KEAP1) to reduce oxidative stress and improve the body’s antioxidant capacity [27]. Na et al. [28] demonstrated in vitro that the stimulation of EGCG on MCF 10A epithelial cells could improve the activity of antioxidant enzymes and promote the release of NRF2, thus improving the antioxidant capacity. Epigallocatechin gallate in green tea can reduce the level of reactive oxygen species (ROS), improve cell viability, inhibit hydrogen peroxide (H_2_O_2_)-induced apoptosis, and thereby reduce the content of MDA [29,30,31]. The addition of polyphenols to the diets of laying hens generally increases the activity of GSH-Px, a finding that was not observed in this study. These results may be related to the different supplemental levels and differences in the animals themselves [32].

The cellulose content of green tea was found to be high in this study. As a substrate for microbial fermentation, cellulose can directly affect the structure of intestinal microorganisms, which in turn can affect the immune regulation of the host, thus increasing the serum levels of IgM, IgG, and IgA [33]. In this experiment, the addition of 0.8% *Green Tea Powder* resulted in increased levels of IgM and IgA in serum. Zeitz et al. [34] found that the supplementation of 0.8% lignocellulose in broiler diets significantly increased the serum IgG and IgA concentrations. This is consistent with the results of the present study.

### 4.3. Egg Quality

The YC is mainly affected by the type and amount of pigments such as carotenoids, carotene, and lutein in the diet, and the addition of exogenous pigments can deepen the YC. Laying hens cannot synthesize these pigments and they must be obtained from the diet [35]. Uuganbayar et al. [36] reported that adding 2% green tea to the diet darkened the YC. In our experiment, the addition of different levels of *Green Tea Powder* made the YC darker, possibly because green tea leaves are rich in pigments and antioxidant substances. Higashiokai et al. [37] identified β-carotene and lutein from the non-polyphenolic parts of green tea. During egg formation, β-carotene and lutein from green tea residue may be deposited in the egg yolk, leading to a darker YC. Green tea is rich in saponins, catechins, flavanols, tannins, phenolic compounds, and other antioxidant substances, which can replace the function of carotenoids, thereby reducing their consumption in the body and depositing more carotenoids in the egg yolk, thus deepening the YC.

The egg’s yolk is important for evaluating the nutritional value of eggs. Fat is accumulated in the egg yolk and is affected by genetic and nutritional factors. In this experiment, the addition of 1.6% *Green Tea Powder* reduced the weight of the egg yolk, which may be related to the tea polyphenols in the *Green Tea Powder*. Kao et al. [38] found that EGCG, an active component of tea polyphenols, can act as a leptin activator, stimulating cell leptin synthesis receptors and increasing the leptin content in the blood of laying hens. Leptin can directly inhibit the synthesis of fat, thus reducing the fat content and the total weight of egg yolk. Muharlien et al. [39] also found that adding 2% green tea to the feed significantly reduced the fat content of the egg yolk.

ET and ES are closely associated with calcium deposition [40]. Some plant extracts have specific functional peptides that can chelate calcium [41]. In our experiment, the addition of *Green Tea Powder* significantly reduced the ET and ES. This may be due to the presence of some special functional peptides in green tea, which chelate calcium in the small intestine and affect the absorption of calcium in the intestine, thus reducing the deposition of calcium carbonate in the eggshell, subsequently reducing the strength and thickness of the eggshell. ET and ES significantly decrease on addition of green tea and *Green Tea Powder* to the diet of laying hens [42,43]. The results are similar to this study. It should be emphasized that ET values were decreased in all experimental groups when compared with the control group.

### 4.4. Amino Acids and Cholesterol in Eggs

The results of this study indicate that dietary *Green Tea Powder* can increase the content of total amino acids, essential amino acids, and umami amino acids in eggs. It was found that adding *Green Tea Powder* increased the dietary CP content, which may be one of the reasons for the increase in the total amino acids in eggs. In addition, phenolic substances in green tea can regulate the metabolic pathway of the body and enhance the genes related to amino acid biosynthesis in the Kyoto Encyclopedia of Genes and Genomes (KEGG) pathway [44]. The content of umami amino acids in eggs may be related to the variety of amino acids in *Green Tea Powder* [45]. Zou et al. [46] found that when tea polyphenols were added to broiler diets, the levels of lysine, glutamic acid, leucine, arginine, and aspartate in the muscle were significantly higher than those in the control group, which was consistent with our results.

### 4.5. Status of Cecal Microflora of Laying Hens

There is mutualistic symbiosis between the host and intestinal microorganisms. Microorganisms participate in the digestion and absorption of host nutrients, helping the host to synthesize vitamins and affecting the regulation of the immune and nervous systems [47,48]. In this experiment, the effective sequence number of the cecal microflora of laying hens supplemented with 1.6% *Green Tea Powder* was significantly higher than that of the CON group. Meanwhile, the PCA showed that the structural distribution of cecal microflora was relatively scattered, with low similarity and substantial differences in community structure, indicating that the addition of *Green Tea Powder* improved the diversity of cecal microflora. Seo et al. [49] reported that fermented tea extract reduced the ratio of *Firmicutes* to *Bacteroidetes* in the large intestine of obese mice. Liu et al. [50] used linear discriminant analysis to identify 30 key intestinal microorganisms such as *Bacteroidetes*, *Trichelicaceae*, and *Rhizobacteria*, which significantly changed after supplementation with tea in the diet. These results are similar to those of the present study.

Many microorganisms live in the intestinal tract of animals, which is considered to be a bioreactor and is the main site for the reaction of polyphenols in green tea with microorganisms [51]. The bioavailability of green tea in animal intestines is low. Undigested phenolic substances interact with colonic microorganisms, and their metabolites affect the number of intestinal microorganisms and the microecological environment, similar to the effects of prebiotics [52]. Prebiotics are mainly oligosaccharides, and EGCG and the fructooligosaccharides in green tea have similar effects on the proliferation of beneficial bacteria such as *Bifidobacterium*. *Bacteroidetes* and *Firmicutes* are the main phyla involved in the metabolism of polyphenols in the intestinal tract. *Firmicutes* are more inhibited by polyphenols and their metabolites, making *Bacteroidetes* in the colon more active [53]. Therefore, in this experiment, compared with the CON group, the addition of green tea residue increased the relative abundance of *Bacteroidetes* at the phylum level but decreased the relative abundance of *Firmicutes*.

Studies on some phenolic compounds have shown that polyphenolic plant extracts inhibit Gram-positive fibrinolytic bacteria and ciliates that degrade fibers [54]. In this experiment, the addition of *Green Tea Powder* increased the abundance of Proteobacteria but did not achieve a significant difference, whereas the abundance of Firmicutes and Heterotrophic bacteria decreased. Proteobacteri*a*, *Firmicutes*, and heterotrophic bacteria are Gram-positive bacteria. These results indicate that phenolic compounds in *Green Tea Powder* have inhibitory effects on Gram-positive bacteria and that adding *Green Tea Powder* in the diet has positive effects on improving the intestinal microbial environment and increasing the microbial diversity, which is beneficial for the intestinal health of hens.

## 5. Conclusions

In conclusion, *Green Tea Powder* is added to the diet of hens during the peak laying period, and this supplementation reduces the EYW, increases the levels of IgM and IgA, and decreases the level of MDA in the serum of hens; reduces the ES and ET of the eggshells and deepens the YC; improves the amino acid content of eggs; and increases the diversity of the intestinal microflora, which improves the structure of the cecal microflora of the laying hens and thus can be beneficial to the health of laying hens.

## Figures and Tables

**Figure 1 animals-14-03020-f001:**
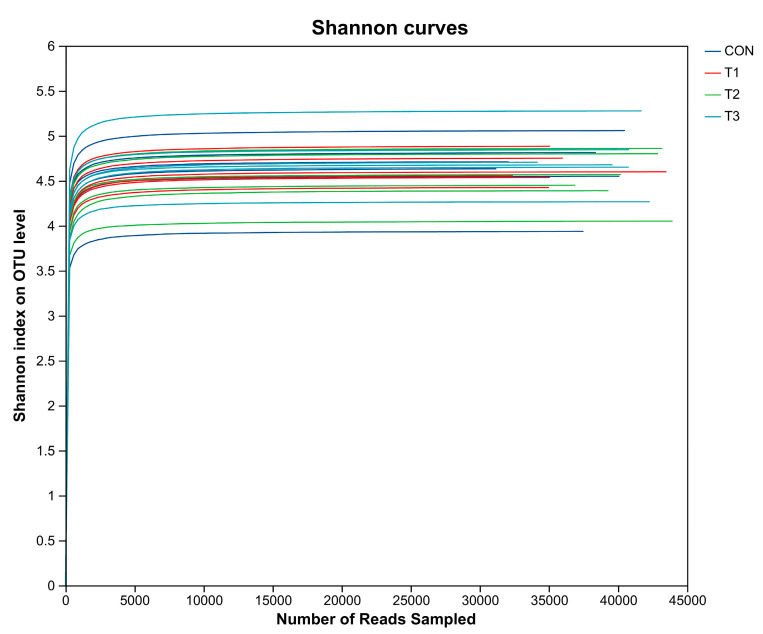
Sequence depth analysis of the cecal flora of laying hens.

**Figure 2 animals-14-03020-f002:**
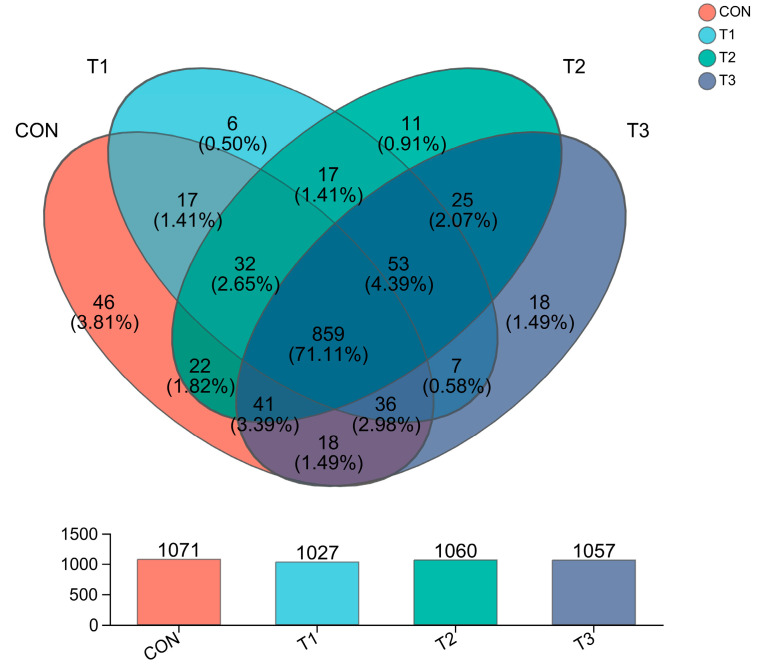
OTU Venn diagram of the cecal flora of laying hens.

**Figure 3 animals-14-03020-f003:**
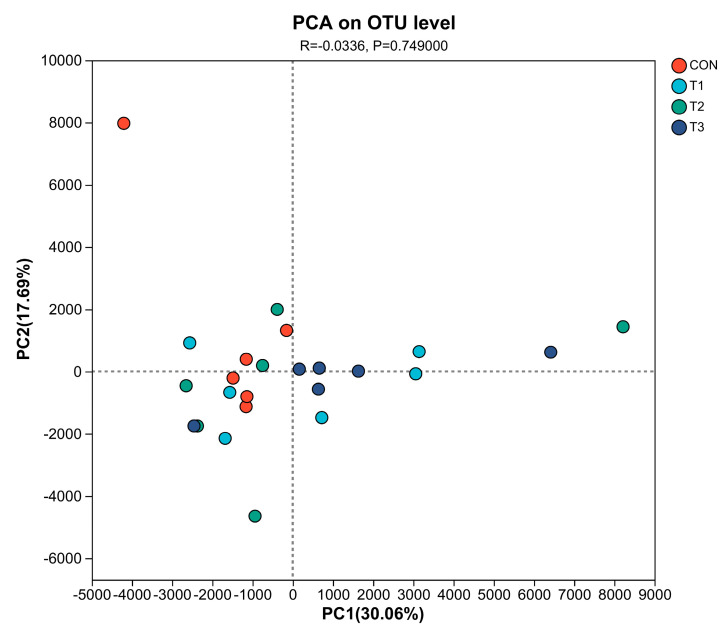
PCA of cecal flora in laying hens.

**Figure 4 animals-14-03020-f004:**
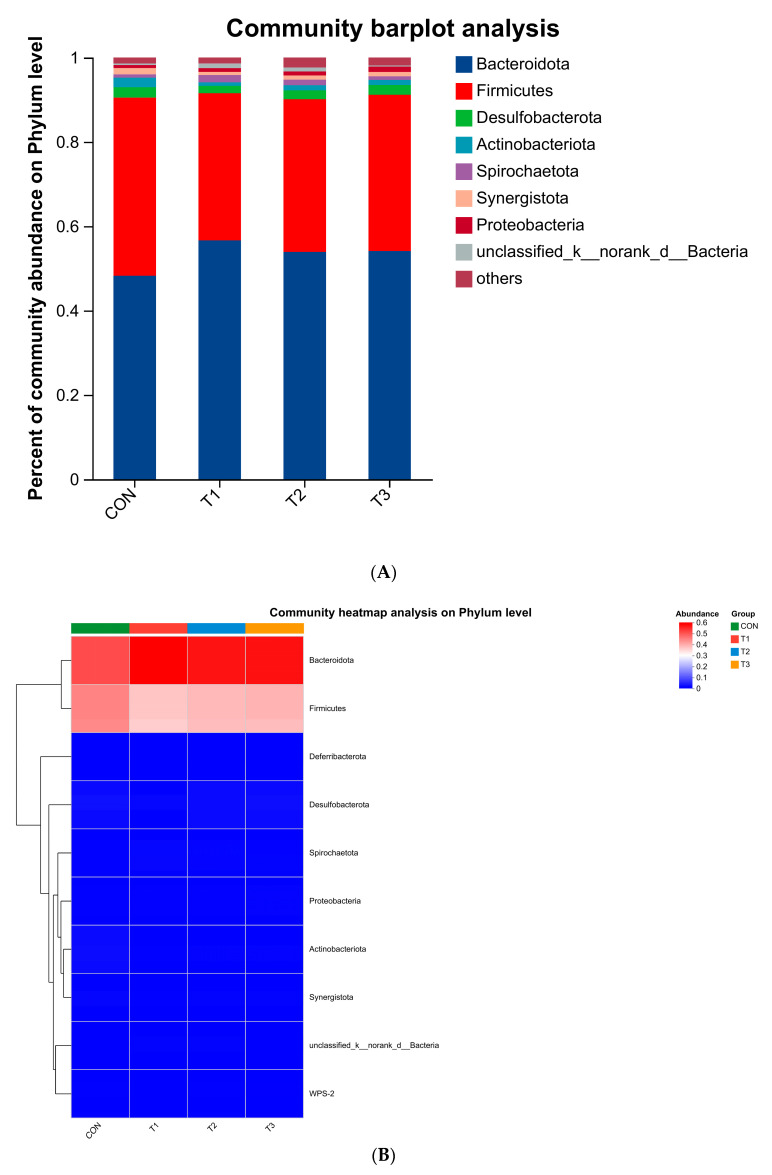
Analysis of relative abundance of species of cecal flora in laying hens at phylum level (**A**). Cluster map of species abundance of cecal flora in laying hens at phylum level (**B**). Analysis of relative abundance of cecal flora in laying hens at family level (**C**). Cluster map of species abundance of cecal flora in laying hens at family level (**D**).

**Table 1 animals-14-03020-t001:** Ingredients and chemical composition of diets with or without *Green Tea Powder* (air-dried basis).

Items	Treatments			
	Control	Trial Group I	Trial Group II	Trial Group III
Ingredients (%)				
Green Tea Powder	0.00	0.80	1.60	2.40
Corn	65.00	64.47	63.95	63.42
Soybean meal	21.00	20.73	20.45	20.18
Wheat bran	3.00	3.00	3.00	3.00
Premix ^1^	2.50	2.50	2.50	2.50
Limestone	3.50	3.50	3.50	3.50
Calcium sand	4.00	4.00	4.00	4.00
CaHPO_4_	1.00	1.00	1.00	1.00
Total	100	100	100	100
Analytical composition (%) ^2^				
DM	95.17	94.82	94.07	93.88
ME (MJ/kg)	17.25	17.15	17.06	16.96
CP	17.24	17.63	17.83	18.08
EE	3.75	2.84	2.29	1.82
Ash	3.46	3.38	3.47	3.20
Ca	1.67	1.85	2.07	2.15
P	0.11	0.14	0.17	0.20
Asp	1.88	1.66	1.65	1.67
Thr	0.75	0.67	0.66	0.66
Ser	0.97	0.86	0.85	0.84
Glu	3.71	3.29	3.24	3.22
Gly	0.83	0.74	0.73	0.73
Ala	1.03	0.93	0.88	0.88
Val	0.93	0.83	0.81	0.81
Met	0.43	0.36	0.40	0.44
Ile	0.82	0.73	0.72	0.73
Leu	1.75	1.57	1.52	1.50
Tyr	0.78	0.68	0.67	0.68
Phe	0.98	0.87	0.85	0.86
His	0.51	0.45	0.45	0.45
Lys	1.09	0.95	0.96	0.98
Arg	1.30	1.15	1.15	1.16
Pro	1.20	1.09	1.04	1.05

^1^ Premix contents (per kg): Vitamin A, 8000 IU; Vitamin D, 1625 IU; Vitamin E, 21.25 IU; Vitamin K_3_, 0.7 mg; Vitamin B_1_, 1.4 mg; Vitamin B_2_, 1.4 mg; Nicotinamide, 20.5 mg; Pantothenic acid, 6.0 mg; Biotin, 0.06 mg; Choline Chloride, 190 mg; Fe, 27.5 mg; Zn, 32.5 mg; Mn, 35 mg; Cu, 5 mg; I, 0.25 mg; Se, 0.065 mg; Ca, 0.35%; TP, 0.08%; NaCl, 0.125%; Met, 0.06%; and moisture, 0.25%. ^2^ Nutrition level is measured value; metabolizable energy is calculated value. DM, dry matter; ME, metabolizable energy; CP, crude protein; EE, ether extract; Ca, calcium; P, phosphorus.

**Table 2 animals-14-03020-t002:** Determination of constituents of *Green Tea Powder* (air-dried basis).

Items	Content (%)
M	8.24
CP	27.61
EE	2.75
FAA	2.33
TP	24.41
CAFF	3.71
GA	0.16
EGC	0.42
C	0.19
EGCG	6.37
EC	1.02
GCG	0.09
ECG	4.80

M, moisture; CP, crude protein; EE, ether extract; FAA, free amino acid; TPs, tea polyphenols; CAFF, caffeine; GA, gallic acid; EGC, epigallocatechin; C, catechins; EGCG, epigallocatechin gallate; EC, epicatechin; GCG, gallocatechin gallate; ECG, epicatechin gallate.

**Table 3 animals-14-03020-t003:** The effect of adding different levels of *Green Tea Powder* to the diet on the growth performance of laying hens.

Items	Groups				*p*-Value
	Control	Trial Group I	Trial Group II	Trial Group III	
ADFI (g)	84.21 ± 3.32	84.11 ± 3.61	86.81 ± 5.31	85.71 ± 8.06	0.438
ADG (g)	2.05 ± 0.25 ^a^	2.27 ± 0.46 ^a^	0.77 ± 0.20 ^b^	0.81 ± 0.21 ^b^	0.010
EP (pieces/d)	0.32 ± 0.04	0.32 ± 0.02	0.34 ± 0.02	0.36 ± 0.04	0.277
FCR	4.63 ± 0.69	4.86 ± 0.28	4.79 ± 0.55	4.50 ± 0.35	0.260

^a,b^ Means with different superscript letters in the same row differ significantly (*p* < 0.05); ADFI, average daily feed intake; ADG, average daily gain; FCR, feed conversion rate; EP, egg production.

**Table 4 animals-14-03020-t004:** The effect of adding different levels of *Green Tea Powder* in the diet on the serum antioxidant and immune indexes of laying hens.

Items	Groups				*p*-Value
	Control	Trial Group I	Trial Group II	Trial Group III	
Serum biochemical parameters					
TP (g/L)	31.59 ± 5.79	32.16 ± 7.93	32.26 ± 5.18	31.54 ± 6.08	0.852
ALB (g/L)	26.55 ± 4.36	23.41 ± 5.33	27.31 ± 4.48	24.23 ± 4.91	0.069
IgA (g/L)	5.50 ± 0.89 ^ab^	6.77 ± 0.69 ^a^	3.64 ± 0.66 ^b^	3.92 ± 0.96 ^b^	0.017
IgG (g/L)	1.34 ± 0.34	1.45 ± 0.12	1.45 ± 0.24	1.60 ± 0.45	0.246
IgM (g/L)	3.79 ± 1.02 ^b^	10.13 ± 3.30 ^a^	4.22 ± 0.77 ^b^	6.15 ± 1.34 ^b^	0.010
Serum redox status					
SOD (U/mL)	132.3 ± 25.51	137.01 ± 18.27	133.92 ± 21.22	124.77 ± 30.44	0.448
MDA(nmoL/mL)	9.05 ± 1.01 ^a^	3.67 ± 0.21 ^b^	4.07 ± 0.32 ^b^	3.13 ± 0.35 ^b^	0.010
GSH-Px (U/mL)	661.95 ± 155.37 ^a^	592.92 ± 79.42 ^a^	435.40 ± 31.11 ^b^	455.75 ± 57.23 ^b^	0.011

^a,b^ Means with different superscript letters in the same row differ significantly (*p* < 0.05); TP, total protein; ALB, albumin; IgA, immunoglobulin A; IgG, immunoglobulin G; IgM, immunoglobulin M; SOD, superoxide dismutase; MDA, malondialdehyde; GSH-Px, glutathione peroxidase.

**Table 5 animals-14-03020-t005:** The effect of adding different levels of *Green Tea Powder* in diets on egg quality.

Items	Groups				*p*-Value
	Control	Trial Group I	Trial Group II	Trial Group III	
AEW (g)	48.56 ± 5.49	48.3 ± 3.61	46.92 ± 3.29	48.69 ± 2.71	0.218
AH (mm)	3.89 ± 0.94	4.16 ± 0.99	3.95 ± 1.07	4.21 ± 0.82	0.372
YC	7.11 ± 1.64 ^b^	7.61 ± 1.75 ^ab^	8.47 ± 1.46 ^a^	7.59 ± 1.21 ^ab^	0.010
HU	62.25 ± 10.35	65.86 ± 9.70	63.74 ± 12.30	65.55 ± 8.94	0.351
EYW (g)	16.84 ± 1.78 ^a^	16.38 ± 1.15 ^ab^	15.65 ± 1.24 ^b^	16.76 ± 2.14 ^a^	0.033
ES (N/m^2)^	35.32 ± 6.80 ^a^	30.65 ± 5.90 ^ab^	26.51 ± 5.88 ^b^	30.74 ± 8.33 ^ab^	0.010
ESI	1.34 ± 0.06	1.37 ± 0.07	1.35 ± 0.01	1.35 ± 0.01	0.256
ET (mm)	0.32 ± 0.02 ^a^	0.30 ± 0.04 ^b^	0.29 ± 0.03 ^b^	0.29 ± 0.04 ^b^	0.030

^a,b^ Means with different superscript letters in the same row differ significantly (*p* < 0.05); AEW, average egg weight; AH, albumen height; YC, yolk color; HU, Haugh unit; EYW, egg yolk weight; ES, eggshell strength; ESI, egg shape index; ET, eggshell thickness.

**Table 6 animals-14-03020-t006:** The effect of adding different levels of *Green Tea Powder* to the diet on the content of cholesterol and amino acids in eggs.

Items	Groups				*p*-Value
	Control	Trial Group I	Trial Group II	Trial Group III	
Cholesterol (mg/100 g)	454.33 ± 31.50	440.00 ± 42.76	471.00 ± 14.11	487.00 ± 34.37	0.069
Amino acid (g/100 g)					
Asp	0.44 ± 0.02 ^b^	0.45 ± 0.01 ^ab^	0.46 ± 0.02 ^ab^	0.47 ± 0.01 ^a^	0.029
Thr	1.30 ± 0.04 ^b^	1.33 ± 0.02 ^ab^	1.34 ± 0.04 ^ab^	1.39 ± 0.04 ^a^	0.012
Ser	0.70 ± 0.03 ^b^	0.72 ± 0.01 ^b^	0.73 ± 0.02 ^ab^	0.76 ± 0.02 ^a^	0.028
Glu	1.81 ± 0.06 ^b^	1.84 ± 0.03 ^ab^	1.87 ± 0.04 ^ab^	1.92 ± 0.01 ^a^	0.014
Gly	0.29 ± 0.01 ^b^	0.29 ± 0.06 ^a^	0.30 ± 0.01 ^ab^	0.31 ± 0.01 ^a^	0.037
Ala	0.52 ± 0.02 ^b^	0.53 ± 0.02 ^b^	0.54 ± 0.02 ^ab^	0.56 ± 0.01 ^a^	0.014
Val	0.51 ± 0.02 ^b^	0.52 ± 0.01 ^ab^	0.52 ± 0.02 ^ab^	0.54 ± 0.01 ^a^	0.010
Met	0.44 ± 0.01	0.44 ± 0.02	0.43 ± 0.01	0.44 ± 0.01	0.290
Ile	0.44 ± 0.02 ^b^	0.45 ± 0.01 ^ab^	0.46 ± 0.01 ^ab^	0.47 ± 0.01 ^a^	0.046
Leu	0.78 ± 0.04 ^b^	0.80 ± 0.02 ^ab^	0.80 ± 0.03 ^ab^	0.83 ± 0.02 ^a^	0.023
Tyr	0.36 ± 0.01	0.36 ± 0.01	0.36 ± 0.02	0.37 ± 0.01	0.217
Phe	0.48 ± 0.02 ^b^	0.49 ± 0.01 ^ab^	0.49 ± 0.01 ^ab^	0.50 ± 0.01 ^a^	0.037
Lys	0.50 ± 0.02 ^b^	0.51 ± 0.01 ^ab^	0.52 ± 0.01 ^ab^	0.53 ± 0.01 ^a^	0.013
His	0.24 ± 0.02	0.23 ± 0.01	0.23 ± 0.01	0.22 ± 0.01	0.125
Arg	0.45 ± 0.02 ^b^	0.46 ± 0.015 ^b^	0.47 ± 0.02 ^ab^	0.50 ± 0.02 ^a^	0.010
Pro	1.09 ± 0.05 ^b^	1.12 ± 0.04 ^b^	1.13 ± 0.02 ^b^	1.55 ± 0.03 ^a^	0.010
TAA	10.36 ± 0.37 ^b^	10.52 ± 0.18 ^b^	10.66 ± 0.26 ^b^	11.36 ± 0.11 ^a^	0.010
EAA	4.45 ± 0.16 ^b^	4.53 ± 0.05 ^ab^	4.56 ± 0.11 ^ab^	4.70 ± 0.10 ^a^	0.023
NEAA	5.91 ± 0.21 ^b^	5.60 ± 0.13 ^b^	6.10 ± 0.15 ^a^	6.67 ± 0.11 ^a^	0.010
EAA/TAA (%)	42.97	43.05	42.79	41.37	
EAA/NEAA (%)	75.34	75.58	74.81	70.57	

^a,b^ Means with different superscript letters in the same row differ significantly (*p* < 0.05); TAA, total amino acid; EAA, essential amino acid; NEAA, nonessential amino acid.

**Table 7 animals-14-03020-t007:** Alpha diversity indexes.

Items	Control	Trial Group I	Trial Group II	Trial Group III	*p*-Value
Valid sequence	36,462 ± 4022 ^b^	36,064 ± 3696 ^b^	40,932 ± 2817 ^a^	39,787 ± 1913 ^ab^	0.022
OTUs	1071 ± 102	1027 ± 82	1060 ± 64	1057 ± 78	0.688
Shannon index	4.62 ± 0.38	4.62 ± 0.16	4.52 ± 0.30	4.74 ± 0.33	0.256
Simpson index	0.03 ± 0.02	0.03 ± 0.01	0.04 ± 0.02	0.03 ± 0.01	0.312
Ace index	756.12 ± 57.90	750.16 ± 21.46	733.32 ± 83.72	767.34 ± 103.17	0.545
Chao1 index	767.91 ± 48.71	759.74 ± 32.61	742.44 ± 74.01	780.52 ± 106.80	0.489
Coverage/%	99.70	99.70	99.70	99.70	0.512

^a,b^ Means with different superscript letters in the same row differ significantly (*p* < 0.05).

**Table 8 animals-14-03020-t008:** The effect of adding *Green Tea Powder* to the diet on the relative abundance of the cecal flora of laying hens at the phylum level.

Items	Groups				*p*-Value
	Control	Trial Group I	Trial Group II	Trial Group III	
*Bacteroidota*	48.46 ± 9.62 ^b^	56.50 ± 8.71 ^a^	53.60 ± 11.45 ^ab^	54.17 ± 6.65 ^ab^	0.037
*Firmicutes*	41.99 ± 10.03 ^a^	34.92 ± 4.37 ^b^	36.49 ± 10.89 ^ab^	37.05 ± 5.75 ^ab^	0.041
*Desulfobacterota*	2.52 ± 0.69	1.75 ± 0.20	2.05 ± 0.57	2.30 ± 0.66	0.595
*Actinobacteriota*	2.28 ± 0.22	0.86 ± 0.08	1.23 ± 0.06	1.19 ± 0.09	0.106
*Spirochaetota*	0.79 ± 0.12	1.75 ± 0.17	1.34 ± 0.38	0.89 ± 0.07	0.390
*Synergistota*	1.50 ± 0.12	0.75 ± 0.09	1.03 ± 0.19	0.98 ± 0.06	0.587
*Proteobacteria*	0.71 ± 0.09	0.89 ± 0.05	0.92 ± 0.09	1.27 ± 0.01	0.219
*unclassified*	0.38 ± 0.05	1.14 ± 0.08	1.06 ± 0.16	0.27 ± 0.09	0.538
*WPS-2*	0.64 ± 0.05	0.35 ± 0.06	0.77 ± 0.05	0.52 ± 0.09	0.637
*Verrucomicrobiota*	0.16 ± 0.03	0.03 ± 0.02	0.12 ± 0.02	0.05 ± 0.01	0.218

^a,b^ Means with different superscript letters in the same row differ significantly (*p* < 0.05).

**Table 9 animals-14-03020-t009:** The effect of adding *Green Tea Powder* to the diet on the relative abundance of the cecal flora of laying hens at the family level.

Items	Groups				*p*-Value
	Control	Trial Group I	Trial Group II	Trial Group III	
*Bacteroidaceae*	17.18 ± 3.98 ^b^	25.05 ± 2.31 ^ab^	24.27 ± 4.96 ^ab^	29.71 ± 7.44 ^a^	0.012
*Rikenellaceae*	18.23 ± 5.32 ^a^	11.16 ± 3.15 ^b^	9.61 ± 1.82 ^b^	8.43 ± 2.21 ^b^	0.016
*Lachnospiraceae*	13.76 ± 3.63	10.09 ± 1.59	11.81 ± 3.37	11.45 ± 1.07	0.487
*Prevotellaceae*	3.30 ± 0.32	7.65 ± 0.15	6.59 ± 1.73	5.84 ± 0.14	0.174
*Ruminococcaceae*	5.25 ± 0.50	4.48 ± 0.27	6.00 ± 0.65	5.09 ± 1.19	0.863
*Acidaminococcaceae*	6.10 ± 0.99	4.22 ± 0.96	4.47 ± 0.62	5.13 ± 0.33	0.753
*Oscillospiraceae*	4.91 ± 0.71	5.05 ± 0.27	4.85 ± 0.01	5.06 ± 0.63	0.994
*Desulfovibrionaceae*	2.52 ± 0.69	1.75 ± 0.20	2.05 ± 0.27	2.30 ± 0.66	0.595
*Tannerellaceae*	1.28 ± 0.76	1.80 ± 0.10	1.59 ± 0.06	2.07 ± 0.89	0.454
*Synergistaceae*	1.50 ± 0.12	0.75 ± 0.09	1.03 ± 0.19	0.98 ± 0.09	0.587
*Spirochaetaceae*	0.70 ± 0.09	1.20 ± 0.06	1.32 ± 0.38	0.87 ± 0.04	0.555
*Butyricicoccaceae*	1.15 ± 0.03	0.87 ± 0.05	0.85 ± 0.10	1.00 ± 0.04	0.750
*Christensenellaceae*	1.53 ± 0.03	0.81 ± 0.03	0.80 ± 0.04	0.56 ± 0.03	0.430
*Lactobacillaceae*	1.12 ± 0.22	1.19 ± 0.22	0.55 ± 0.09	0.84 ± 0.04	0.759
*Atopobiaceae*	1.24 ± 0.16	0.44 ± 0.06	0.82 ± 0.01	0.63 ± 0.05	0.490
*Peptostreptococcaceae*	0.35 ± 0.06	0.51 ± 0.01	0.96 ± 0.09	0.30 ± 0.02	0.740

^a,b^ Means with different superscript letters in the same row differ significantly (*p* < 0.05).

## Data Availability

All data presented in this article are available upon request to the author if necessary.

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
