# Peer review of "Effects of Different Levels of Green Tea Powder on Performance, Antioxidant Activity, Egg Mass, Quality, and Cecal Microflora of Chickens"

_animals, 2024, doi:10.3390/ani14203020_

Round 1

Reviewer 1 Report

Comments and Suggestions for Authors

The present study aimed to investigate the effects of various levels of green tea powder supplementation on the production performance, egg quality, serum immune indices, antioxidant indices, and caecum microflora of laying hens. The study is well-designed, and data collection is appropriate. Few suggestions/changes are required to further improve the manuscript.

1. In title, please the “Antioxidant Aactivity” as “Antioxidant Activity”.

2. In simple summary, please rephrase this sentence “Increase the amino acid content of eggs”.

3. In simple summary, please rewrite this sentence as it seems to be the objectives which is already mentioned at the start of the summary “This study investigated the effects of Green Tea Powder supplementation on the performance and egg quality of laying hens, and….”.

4. In the abstract, please remove these phrases as they are important to add in this section “Each hen was weighed on the first and last day of the experiment, and feed intake was recorded daily during the experiment. On the last day of the experiment, 36 eggs from each group were collected for analyzing the egg quality and amino acid content. Two hens per replicate were slaughtered, and the serum and cecum contents were analyzed”.

5. In the abstract, please remove the numbering of the results.

6. In the abstract, please remove the abbreviations used like ADG, MDA, IgM, IgA, EYW etc. as they are not used again in the abstract.

7. In the abstract, please rewrite the conclusion by specifying the dose effects on specific parameters that are affected by green tea powder supplementation rather than mentioning the comprehensive effect.

8. In the introduction, the authors wrote whole paragraph with only one citation which is not according to the writing ethics. Please add proper citation/references after one or two phrases.

9. In the materials and methods, please remove these sentences are this is not the necessary information “The detailed procedure is as follows: Chicken thigh lateral thigh muscles were injected with 150 mg/kg sodium pentobarbital to euthanasia, and then the cecal contents were dissected and collected for cecal microbiota determination”.

10. In the materials and methods, please rewrite this sentence “Each group consisted of 6 replicates with 15 hens replicate” as “Each group consisted of 6 replicates with 15 hens per replicate”.

11. In the materials and methods section, “Each replicate was further divided into five cages, with three laying hens in each cage….” The authors mentioned that at the end of the experiment, two birds from each replicate were selected for sampling means twelve birds per group, how the bird’s selection is carried from the five cages within the replicate to ensure randomization?

12. In the materials and methods section, while explaining the procedure of parameters, please make them brief and avoid numbering. Moreover, it is recommended to cite the already reported procedures with a very brief explanation of any modification made by the authors in the procedure.

13. In the statistical analysis, “Data were presented as “mean ± standard deviation (SD).” Please confirm whether it is mean ± standard deviation or mean ± standard error of mean (SEM) because the data in the tables seems to be presented as mean ± SEM.

14. In the statistical analysis, as the dose-dependent response of green tea powder is studied it is suggested to apply polynomial contrasts to check the linear or quadratic effects of various supplemented doses.

15. In the results section, please also write the non-significant results. Also, please re-write the results by clear comparison with the control.

16. In table 4, the data presented under the heading of serum antioxidant index includes SOD, MDA, and GSH-Px. In these three parameters, SOD and GSH-Px are antioxidant enzymes while the MDA is an oxidative stress marker. It is suggested to replace the title “serum antioxidant status” with “serum redox status” which will cover both the oxidant and antioxidant attributes.

17. The redox status shows no effect on SOD and decreased GSH-Px levels in layers supplemented with 1.6% and 2.4 % green tea powder compared with the control group. Then how the authors claim that green tea powder exhibits antioxidant potential?

Reviewer 2 Report

Comments and Suggestions for Authors

Suggestions for Improvement:

Abstract: It is advisable to include a concise overview of the statistical analyses performed. This addition will clarify the methods employed and improve the overall transparency of the research.

Introduction: Although China is cited as the leading producer, it would be beneficial to highlight the wider implications of this study. Consider elaborating on how these findings could assist other nations in producing similar products and broadening economic prospects on a global scale.

Methodology: A well-defined justification for incorporating Green tea powder as a substitute for corn and soybean meal should be presented. Since this change impacts the diet's composition and mixture, detailing its effects on nutritional balance and performance is crucial. Furthermore, explaining why the diets were not modified in nutrient content to uphold nutritional quality would further reinforce the methodology.

Feed Conversion Ratio (FCR): Elucidate how the FCR was derived, especially since the figures seem elevated. Could this be associated with egg production? Providing clarity on this matter would enrich the context surrounding the results.

Figures: The clarity of the cecal microbiota graphs needs to be enhanced to ensure they are easily readable. This improvement will bolster the visual depiction of the data and aid the reader in interpreting the findings.

Discussion: This section should incorporate more specific data that underscores the wider implications of the results. A thorough summary of the study’s outcomes, which synthesizes the findings and their possible impacts, could be included at the conclusion of the discussion, just prior to the final remarks. This would deliver a cohesive summary of the study’s essential points and potential applications.

Conclusion: To enhance the effectiveness of the conclusion, modify it to reflect the present tense. This change will project confidence in the findings and distinguish the conclusion from the discussion. Strong, assertive statements are particularly crucial in this section.

Comments on the Quality of English Language

I would suggest conducting a comprehensive review of the English language throughout the manuscript to enhance readability and overall understanding. Some sentences are intricate and could be made simpler for better clarity, while certain technical terms or phrases might benefit from rewording to ensure that the intended meaning is easily grasped by a wider audience. A professional English editing service could also assist in improving grammar, punctuation, and sentence structure, which would contribute to the manuscript’s clarity and flow.

Round 2

Reviewer 1 Report

Comments and Suggestions for Authors

All the suggestions/comments are addressed by the authors appropriately.